# Bio-inspired Murray materials for mass transfer and activity

Xianfeng Zheng[1,†], Guofang Shen[1], Chao Wang[1], Yu Li[1], Darren Dunphy[2], Tawfique Hasan[3], C. Jeffrey Brinker[2,4] & Bao-Lian Su[1,5,6]

Both plants and animals possess analogous tissues containing hierarchical networks of pores, with pore size ratios that have evolved to maximize mass transport and rates of reactions. The underlying physical principles of this optimized hierarchical design are embodied in Murray's law. However, we are yet to realize the benefit of mimicking nature's Murray networks in synthetic materials due to the challenges in fabricating vascularized structures. Here we emulate optimum natural systems following Murray's law using a bottom-up approach. Such bio-inspired materials, whose pore sizes decrease across multiple scales and finally terminate in size-invariant units like plant stems, leaf veins and vascular and respiratory systems provide hierarchical branching and precise diameter ratios for connecting multi-scale pores from macro to micro levels. Our Murray material mimics enable highly enhanced mass exchange and transfer in liquid–solid, gas–solid and electrochemical reactions and exhibit enhanced performance in photocatalysis, gas sensing and as Li-ion battery electrodes.

[1] State Key Laboratory of Advanced Technology for Materials Synthesis and Processing, Wuhan University of Technology, Loushi Road 122, Wuhan 430070, China. [2] NSF/UNM Center for Micro-Engineered Materials, Department of Chemical and Nuclear Engineering, The University of New Mexico, Albuquerque, New Mexico 87131, USA. [3] Cambridge Graphene Centre, Department of Engineering, University of Cambridge, Cambridge CB3 0FA, UK. [4] Advanced Materials Lab, Sandia National Laboratories, 1001 University Boulevard SE, Albuquerque, New Mexico 87106, USA. [5] Laboratory of Inorganic Materials Chemistry, Department of Chemistry, University of Namur, 61 rue de Bruxelles, B-5000 Namur, Belgium. [6] Clare Hall, University of Cambridge, Herschel Road, Cambridge CB3 9AL, UK. † Present address: School of Chemical Engineering and AIBN, The University of Queensland, St Lucia, Brisbane, Queensland 4072, Australia. Correspondence and requests for materials should be addressed to B.-L.S. (email: bao-lian.su@unamur.be or to L.Y. (email: yu.li@whut.edu.cn).

Natural systems and their hierarchical organization are not only optimized and designed for durability but also have the capability to adapt to their external environment, to undergo self-repair, and to perform many highly complex functions[1–4]. To achieve transfer and exchange of substances with extremely high efficiency and minimum energy consumption, evolution by natural selection has endowed many classes of organisms with hierarchically porous networks, in which the pore sizes regularly decrease (and branch) across multiple scales and finally terminate in size-invariant units, such as those seen in plant stems, leaf veins and vascular and respiratory systems[5,6]. The entire natural porous network connected within a finite volume minimizes transport resistance for all the pores and ensures fluent transfer throughout the network as a precondition, commonly referred to as Murray's law[7–10]. By further branching and space-filling to maximize the exchange surface, such networks have to be initiated with coarse macropores (>50 nm), extended by multi-scale pores with increasing numbers but reducing diameters[11–13]. The organisms, described as living Murray networks, thus sustain life and grow obeying Murray's regularity with precise diameter ratios for connecting multi-scale pores from macroscopic to microscopic levels. For example, in plant stems and leaf veins, the sum of the radii cubed, that is, the pore volume of pores, remains constant across every branch point to maximize the flow conductance, which is proportional to the rate of photosynthesis[8,11]. For insects relying upon gas diffusion for breathing, the sum of radii squared, that is, the surface area of tracheal pores remains constant along the diffusion pathway, to maximize the delivery of $CO_2$ and $O_2$ in gaseous forms, about $10^4$ and $10^6$ times faster in air than in water or tissues[8,12,13]. Indeed, Murray's law stipulates the optimized hierarchical design for porous materials with maximized transfer properties[7–10].

Human progress has long benefitted from natural and man-made materials inspired from nature's hierarchical structures[1–4,7–10]. Since its discovery as the basis for vascular and blood systems, Murray's law has attracted very little attention and has thus far been completely overlooked in the areas of physics, chemistry and applied materials. By mimicking nature's hierarchical networks in synthetic materials with multi-scale pores based on Murray's law, such synthetic Murray materials can potentially offer important structural superiority and performance enhancement for a wide range of applications such as in photocatalysis, gas sensing and Li-ion batteries, which could help mitigating current energy and environmental issues[14–19]. It is envisioned that the introduction of Murray's law into the design of materials could revolutionize properties of engineered materials and launch a new era in the field of materials. However, the original Murray's law is only applicable to mass transfer processes involving no mass variations[8,9]. Significant theoretical advances need to be made to apply Murray's principle more broadly to the fields of chemistry, applied materials and industrial reactions.

The way forward towards its applications also faces bottlenecks in the construction of multi-scale interconnected pores. Recently, various synthesis methods have been developed for the preparation of hierarchically porous materials[4]. The most popular approach to fabricate macropores (>50 nm) uses self-assembled silica or polymer spheres as a starting template[14,15]. However, in these strategies, removal of sacrificial templates by washing with hydrofluoric acid or calcination at high temperature (for example, at 500 °C under oxygen atmosphere) severely restricts the choice of materials and thus, the functionality[14,15]. This is because most of the functional metal oxides and metal substrates ideal for applications in energy, sensing and so on are unstable under such harsh chemical treatments. Further crystallization of these materials via high-temperature annealing is usually necessary, but results in clogged mesopores (2–50 nm) or micropores (<2 nm). Taking inspiration from nature's self-organization of cellular units into complex organisms, self-assembly of nanocrystals as building blocks may offer a very broad and applicable approach towards the development of porous materials[16,17]. Nevertheless, progress in this area remains at an elementary level, hindered by the lack of control towards the design of complex hierarchical structures[17–19].

Here we demonstrate a bio-inspired, self-assembled material with space-filling macro–meso–micropores (M–M–M) designed based on revisited Murray's law. The hierarchically porous networks are formed using a bottom-up, layer-by-layer evaporation-driven self-assembly process employing microporous nanocrystals as the primary building blocks under ambient conditions. Such porous Murray materials, composed of interconnected channels with precise dimensions spanning the macro-, meso- and micro-length scales can be fabricated for a broad range of applications. In this work, we demonstrate three different representative and important applications of the same Murray material in the energy and environmental fields. Due to highly enhanced mass exchange and transfer in liquid–solid, gas–solid and electrochemical reactions, we achieve outstanding performance compared to state-of-the art materials in the areas of photocatalysis, gas sensing and Li-storage, respectively. Our work on designing materials according to Murray's law paves the way for pursuing optimized properties of hierarchically porous materials for various applications.

## Results

**Murray's law for vascularized networks.** Over millions of years of evolution, biological organisms have developed a highly hierarchical structure obeying Murray's law for optimized transfer and exchange performance. For example, to optimize the rate of photosynthesis for energy conversion, plants possess leaf veins that show increasing number of branches and narrowing porous channels from macro to micro levels for water and nutrient transfer (Fig. 1a)[11,20]. The SEM image of the cross-section of leaf vein reveals layer-by-layer macroporous networks for such efficient transportation (Fig. 1b). Similarly, for efficient breathing, insects employ open spiracles with hierarchical pores following Murray's law for gas diffusion (Fig. 1c)[12,13,21]. A representative SEM image of spiracles (Fig. 1d) from an insect also shows open macropores on the body surface, which are extended in the interior to enable gas diffusion in and out of the tissues efficiently[22]. Figure 1e,f presents the hierarchically porous network models abstracted from these living Murray systems. These include a parent circular pore (radius $r_0$) connecting to many children pores (radius $r_i$) for substance transfer and exchange. According to the original Murray's law involving no mass variations (see Methods and Supplementary Methods), it can be theoretically deduced and experimentally verified that for laminar flow transfer in plant stems and leaf veins, the sum of the radii cubed of pores remains constant (Fig. 1e)[8,11]. On the other hand, for insect breathing systems, Murray's law dictates an area-preserving branching network formation for optimized gas diffusion (Fig. 1f)[8,12,13].

Original Murray's law has also been used for theoretical prediction of optimum mass diffusion in gaseous or liquid phase, and also for ion transfer[8–10]. However, since its discovery, little attention has been paid to exploit this law for designing advanced materials, reactors and industrial processes for maximizing mass transfer to improve material performance and process efficiency[14–19].

Following Murray's law, we emulate optimized natural systems on the basis of a bottom-up layer-by-layer evaporation-driven

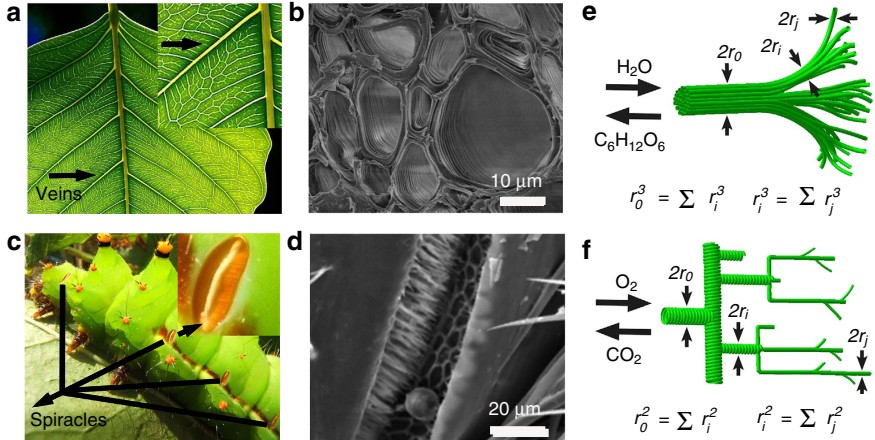

**Figure 1 | Hierarchically porous structures of living Murray networks in leaf and insect.** (**a**) Photographic image of leaf veins. (**b**) SEM image of leaf veins. (**c**) Photographic image of spiracles from an insect. Reproduced from ref. 21 (CC-BY-SA, Wikimedia Commons). (**d**) SEM image of spiracles from an insect. Reproduced from ref. 22 (CC-BY-SA, Wikimedia Commons). Hierarchically porous network models for the (**e**) veins and (**f**) spiracles.

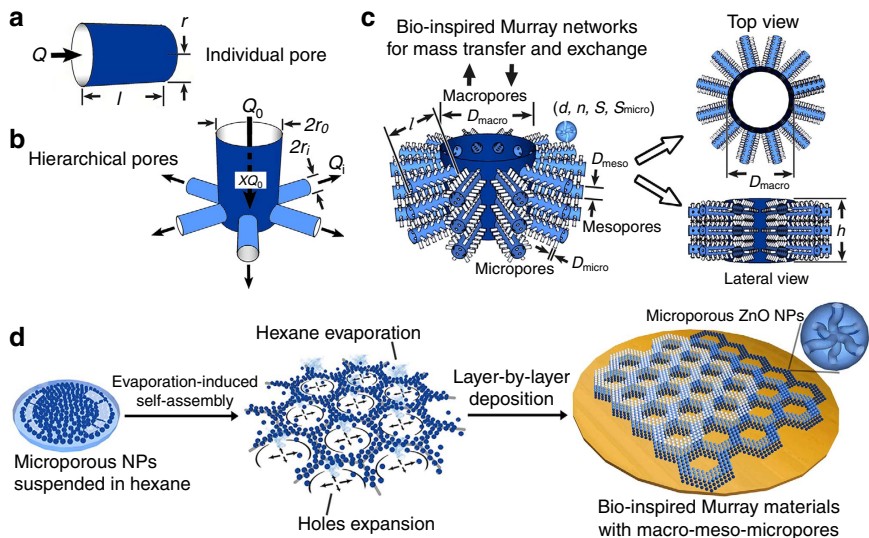

**Figure 2 | Hierarchically porous networks in Murray materials via self-assembly with macro–meso–micropores.** (**a**) Individual pore model. (**b**) Hierarchical pore model with a parent pipe connecting with many children pipes. (**c**) Hierarchically porous network model abstracted from our hierarchically macro–meso–microporous materials: $d$ is the diameter of nanoparticles (NPs), $n$ is the average number of micropores within a single nanoparticle, $S$ is the specific surface area of all the NPs, $S_{micro}$ is the surface area of the micropores, $D_{micro}$ and $D_{meso}$ are the diameters of micropores and mesopores, respectively. (**d**) Fabrication process of the Murray materials via layer-by-layer evaporation-driven self-assembly of NPs.

self-assembly (EDSA) strategy. Well-defined microporous ZnO nanoparticles are used as the primary building blocks, which are further assembled into vascularized Murray networks with interconnected macropores, mesopores and micropores.

Figure 2 illustrates the porous network model abstracted from our desired hierarchically porous material self-assembled by NPs. Murray's principle is based on optimizing mass transfer by minimizing transport resistance in pores with a given total volume. We revisited original Murray's law and developed the generalized Murray's law applicable to optimized mass transfer involving mass variations (see Methods and Supplementary Methods).

As shown in Fig. 2a,b, we use Murray's principle to optimize mass transfer for an individual circular pore and for hierarchical pores with a parent pore (radius $r_0$) and many children pores (radius $r_i$; see Methods). By combining Murray's principle and the law of mass conservation, we deduced the generalized Murray's law involving mass variations (mass loss/increase ratio

$X$) in synthetic vascularized Murray materials: $r_0^\alpha = \frac{1}{1-X} \sum_{i=1}^{N} r_i^\alpha$, where the exponent $\alpha$ (2 or 3) is dependent on the type of the transfer, and $X$ is the ratio of mass variation during mass transfer in the parent pore. This demands a reasonable tolerance adjusted for area-preserving branching networks involving mass diffusion or ion transfer ($\alpha = 2$) with mass variations during the optimum transfer process (Methods). The generalized Murray's law is applicable for optimizing mass transfer in chemical reactions and other mass diffusion or ionic transfer processes involving mass variations.

For the case of our materials with space-filling pores for mass transfer and exchange (Fig. 2c), Murray's law demands hierarchical branching networks with well-defined regular and multi-scale pores to achieve optimum transfer along the pathway. The macroporous channels are openings, similar to those found in the leaf veins and insect spiracles, and are further connected to numerous surrounding mesoporous branches. To maximize mass exchange, organisms pursue full-branching but with narrowing

pores, depending on the size restriction of the functional cells. Taking such biological restrictions into account, microporous branching connected with the upper mesopores can be further introduced through porous nanoparticles. For microporous NPs self-assembled on a substrate surface, the size of the NPs and the film thickness have a finite value, thus leading to a given pore volume. To connect macropores with mesopores, and mesopores with micropores, we applied the generalized Murray's law to derive the size ratios between multi-scale pores expressed by equations (1) and (2) (see Methods).

$$h = \frac{d^2}{\pi D_{meso}^2} D_{macro} \quad (1)$$

$$l = \frac{(1 - X)dD_{meso}^2}{nD_{micro}^2} = \frac{S_{micro}}{S} \frac{dD_{meso}^2}{nD_{micro}^2} \quad (2)$$

The physical relationships in the equations can be used as a guide for the fabrication of Murray materials. These equations reveal that the micro-scale parameters for materials (Fig. 2c), including the film thickness ($h$), wall width ($l$) and diameter ($D_{macro}$) of the macropore are closely dependent on the nanoscale structural parameters of the NP building-blocks ($d$, $n$, $S$, $S_{micro}$, $D_{micro}$ and $D_{meso}$). Here, $d$ is the diameter of nanoparticles, $n$ is the average number of micropores within a single NP, $S$ is the specific surface area of all the NPs, $S_{micro}$ is the surface area of the micropores $D_{micro}$ and $D_{meso}$ are the diameters of micropores and mesopores, respectively.

**Fabrication of Murray materials**. To fabricate synthetic vascularized Murray materials, uniform microporous nanoparticles with defined microporosity were first synthesized as the primary building unit using a controllable organic solution-phase approach (*vide infra*). The secondary and tertiary mesoporous and macroporous networks were constructed by our bottom-up layer-by-layer evaporation-driven self-assembly procedure under room temperature and atmospheric pressure conditions. The solvent evaporation drove the formation of uniform networks of macropores and assembled mesoporous arrays of the primary microporous nanoparticles, resulting in a hierarchical porous network where the pore size ratio was defined by the generalized Murray principle. Layer-by-layer deposition of pre-fabricated Murray networks extended the structures in the thickness direction defining the length of the macropores.

As an example, we employed multi-functional ZnO NPs to construct the Murray material with M–M–M (Fig. 2d). This is because ZnO is a widely used multi-functional material in various applications. A Zn-oleylamine complex prepared at 80 °C is thermally decomposed at 150 °C to form ZnO NPs[23,24]. During the reaction, spontaneous release of gas molecules from the precursors leads to the formation of nanochannels within the ZnO NPs[25–27]. After tuning the ratio of the reactants and the reaction time, uniform ZnO nanocrystals of 30 nm in diameter ($d = 30$ nm) are obtained. Each NP contains on average eight microporous channels ($n = 8$), confirmed via transmission electron microscopy (TEM) (Supplementary Figs 1, 2 and 3). Argon adsorption–desorption measurements of the self-assembled pristine NPs (micro and mesoporous) allowed determination of the specific surface areas for all NPs in a sample ($S = 76$ m$^2$ g$^{-1}$, $S_{micro} = 38$ m$^2$ g$^{-1}$), and the diameters of micropores and mesopores ($D_{micro} = 1.1$ nm, $D_{meso} = 18$ nm) (Supplementary Table 1). For the construction of a single layer of macro–meso–microporous Murray network using microporous ZnO NPs with the above parameters, the macroporous wall width $l$ should be ~502 nm according to equation (2). Evaporation-driven self-assembly of the NPs

dispersed in hexane was used to obtain the macro–meso–microporous film with the required macroporous structures on various substrates. Holes induced by the spontaneous evaporation of hexane could nucleate and expand, forming macroporous rings[28,29]. The NPs are pushed to the edges and aggregated by the expanding holes, resulting in compact walls composed of mesopores between the NPs[30]. For self-assembling 30 nm microporous ZnO NPs at room temperature, we find that the distribution density of vapour holes and their expansion range are strongly influenced by the ZnO concentration in hexane, varying from 0.03125 to 0.5 mg ml$^{-1}$ (Supplementary Fig. 4). Meanwhile, the inter-particle mesoporosity remains invariable during the EDSA process. After tuning the ZnO concentration, a layer of Murray material with a well-defined macroporous network and wall width $l$ of ~0.5 µm and macropore diameter $D_{macro}$ of ~1 µm is obtained. For building a three-dimensional (3D) Murray macro–meso–microporous network with the above parameters following equation (1), the required film thickness is $h$ ~885 nm. This could be achieved using a layer-by-layer deposition technique (Fig. 2d).

Figure 3a shows a macro–meso–microporous network realized via layer-by-layer evaporation-driven self-assembly of ZnO NPs from a 0.25 mg ml$^{-1}$ suspension in hexane on a carbon-coated TEM grid (inset of Fig. 3a). A large area with open ~1 µm diameter macropores is clearly observed, accompanied by many net-like walls along the rims, with an average width of ~0.5 µm ($l$) (Fig. 3b). Under higher magnification, these compact walls are found to be constructed from numerous closely packed NPs, resulting in an inter-particle mesoporous space of ~18 nm ($D_{meso}$) (Fig. 3c). Moreover, numerous open micropores in the NPs are observed with ~1 nm pore diameter ($D_{micro}$; Fig. 3d). A high-resolution TEM lattice image shows the crystallized pores and lattice fringes, with an adjacent fringe spacing of 1.91 Å (inset in Fig. 3d). This corresponds to the (102) planes for hcp ZnO, in agreement with the XRD data analysis. TEM observation confirms that this hierarchical macro–meso–microporosity is topographically highly interconnected throughout one solid body, resulting in fully open macroporous networks branched by surrounding mesoporous channels, which are further extended by the internal micropores in NPs. The synthesized material containing porous channels with increasing numbers and reducing diameters shows bio-mimetic space-filling fractal-like pores following Murray's Law, which are expected to endow the structure with greatly improved mass transfer and exchange. Figure 3e shows a 3D vascularized porous network with a ~0.9 µm thickness, after layer-by-layer deposition on a Si wafer (ZnO M–M–M, Methods). The net-like walls with ~0.5 µm width are constructed with numerous ZnO NPs and exhibit a layered structure (Fig. 3f). Materials with bio-inspired structure and Murray regularity were also self-assembled via layer-by-layer deposition on to Cu foils (Fig. 3g–i) with a film thickness of ~0.9 µm. Large-area open macropores with ~1 µm diameter are also demonstrated in these samples under SEM observation.

Three-level Murray regularity of the synthesized ZnO structure with M–M–M is further demonstrated by mercury porosimetry and argon adsorption. Mercury intrusion measurements confirmed a macropore distribution centred at 1 µm ($D_{macro}$; Fig. 3j). Argon adsorption–desorption isotherms analysed via the Barrett–Joyner–Halenda model[26,27] reveal a narrow mesopore distribution with a mean size of 18 nm ($D_{meso}$) (Fig. 3k), corresponding to the inter-particle space of 3D closely packed microporous NPs. This spacing is consistent with that determined from grazing-incidence small-angle x-ray scattering measurements (Supplementary Fig. 5). Micropores with ~1.1 nm size ($D_{micro}$) within the NPs are elucidated from the low relative pressure region of the argon adsorption isotherm using the

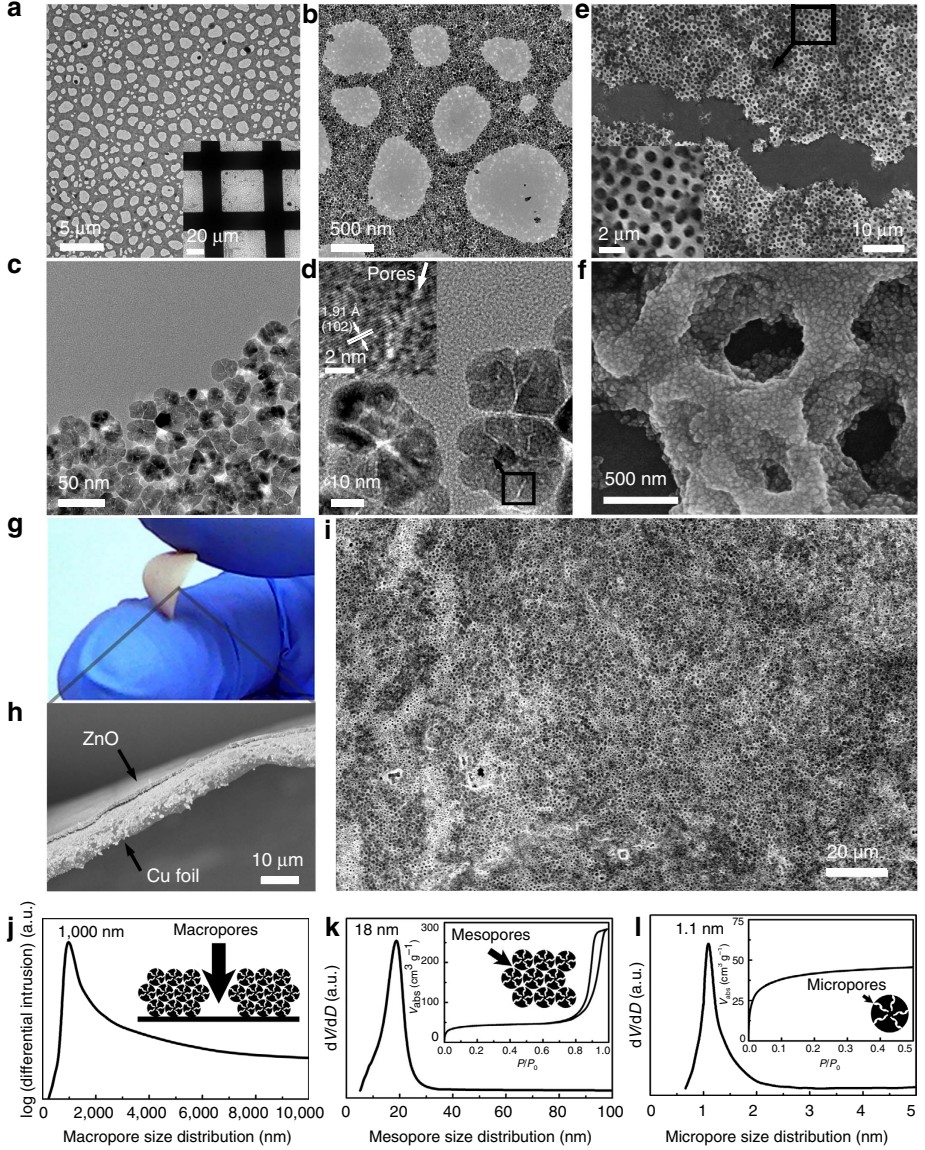

**Figure 3 | Hierarchical structures with macro–meso–micropores (M–M–M) self-assembled by microporous ZnO NPs.** (a–d) TEM images with various magnifications showing the (a) macropores, (b,c) mesopores and (d) micropores arising from the layer-by-layer evaporation-driven self-assembly of NPs in a hexane suspension with 0.25 mg ml$^{-1}$ concentration. The inset shows the high-resolution lattice images of the marked area. SEM images with (e) low and (f) high magnification of self-assembled ZnO M–M–M on Si wafer. (g) Photographic image, (h) SEM cross-section image and (i) SEM image of ZnO M–M–M on Cu foil. Pore size distribution curves for (j) macropres, (k) mesopores and (l) micropores from ZnO M–M–M. The insets in k,l show the corresponding Ar adsorption–desorption isotherms.

non-local density functional theory method[31] (Fig. 3l). The pore-size measurements are based on the model of circular porous channels, consistent with the pore model in Murray's law[8,9]. The pore characterization results confirm that the pore size ratios for macro–mesopores and meso–micropores are very close to Murray's regularity in equations (1) and (2). Thus, the synthetic Murray material is expected to greatly favour mass transfer for applications where mass diffusion or ion transfer is the dominant transport process[8–10,18,19]. This is demonstrated in the following section, through example applications in photocatalysis, gas sensing and Li-storage. ZnO with meso–micropores (M–M) built by identical microporous NPs and ∼19 nm mesopores, and ZnO with only mesopores (M) built by 30 nm solid NPs and mesopores of 22 nm, and purchased bulk ZnO (Methods, Supplementary Methods, Supplementary Table 1 and Supplementary Figs 6 and 7) are used as standards for performance comparisons.

**Application examples**. Figure 4a presents the photocatalytic efficiencies of different ZnO samples for the degradation of rhodamine B (RhB) under ultraviolet irradiation (Supplementary Fig. 8). When the ZnO NPs submerged in an aqueous solution of RhB are irradiated by ultraviolet light, they absorb photons and react with the $H_2O$ molecules. This produces hydroxyl radical species $^{\bullet}OH$ which decompose RhB[32–34]. The diffusion of RhB and $^{\bullet}OH$ with reaction products (through the porous networks) is a dominant transfer process that influences the degradation rate[34]. The rates can be obtained from the slope of plots of $-\ln(C/C_0)$ versus irradiation time, since the photocatalytic degradation profiles are consistent with a pseudo-first-order kinetic model[34,35]. With the Murray network architecture for mass diffusion and exchange, our vascularized micro–meso–macroporous ZnO (ZnO M–M–M) shows an extremely high degradation rate: 2.5 times higher than that of micro–mesoporous ZnO (ZnO M–M) containing two level Murray regularity, 5 times

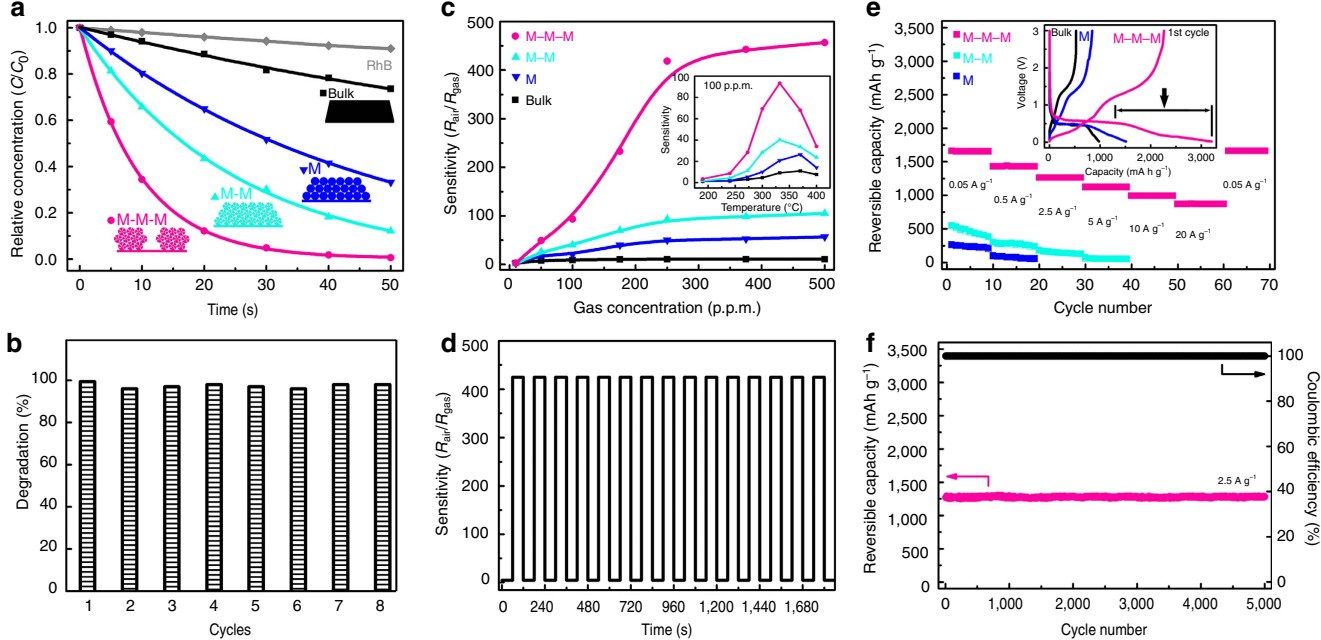

**Figure 4 | Performance of bio-inspired Murray materials.** (**a**) Rhodamine B (RhB) photodegradation rates of different ZnO samples. (**b**) Repeated photodegradation reactions using ZnO with macro–meso–micropores (M–M–M). (**c**) Sensitivities of different ZnO samples exposed to ethanol vapour of various concentrations at 330 °C. The inset shows the sensitivities of different samples exposed to 100 p.p.m. ethanol vapour at different temperatures. (**d**) Repeated response for ZnO M–M–M sample to 250 p.p.m. ethanol at 330 °C. (**e**) Rate capabilities under increasing current densities for ZnO samples after 50 cycles. The inset shows the voltage profiles during the 1st cycle with a current density of 0.05 A g$^{-1}$. (**f**) Long-term cycling performance with a current density of 2.5 A g$^{-1}$ for ZnO M–M–M sample after rate measurement.

higher than that of ZnO containing only mesoporosity (ZnO M), and 17 times higher than that of the ZnO bulk sample. Compared to ZnO M–M with an equivalent specific surface area (76 m$^2$ g$^{-1}$), the macropores connecting to mesopores with Murray regularity in ZnO M–M–M increase the degradation rate by 2.5 times. This can be attributed to a boosted mass transfer in macro–meso–microporous Murray networks[35]. Highly enhanced accessibility of diffusing species to the films were further confirmed by monitoring the dark adsorption kinetics of RhB in agreement with a pseudo-first-order model[36]. This reveals that ZnO M–M–M delivered a 2.5 times faster adsorption rate of RhB than ZnO M–M (Supplementary Fig. 9). Photocatalysis, inspired by plants for photosynthesis with Murray networks, has widely been used for degrading organic pollutants based on fully dispersed nano-semiconductors[37,38]. Using a 3D Murray network inspired from photosynthetic leaves, the relative photocatalytic rate of ZnO M–M–M is comparable to that of fully dispersed ZnO NPs (~25 nm) via stirring in a RhB solution with the best photocatalytic activity ever reported for ZnO nanomaterials[32,38]. In addition to being highly efficient, our ZnO M–M–M film is also easily recyclable for repeated reactions (Fig. 4b).

The gas-sensing performance of our ZnO M–M–M Murray material under ethanol vapour exposure is demonstrated in Fig. 4c. When the ZnO NPs are exposed to air, adsorption of oxygen molecules on their surfaces causes an increase in electrical resistance ($R_{air}$)[18,39]. After the diffusion of ethanol vapours and reaction with oxygen molecules, the electrical resistance ($R_{gas}$) decreases. We define the sensitivity of the sensor as $R_{air}/R_{gas}$. The diffusion of ethanol, accompanied by the production of $CO_2$ via reaction with adsorbed oxygen on the porous film is a dominant transfer process that influences the gas-sensing performance[18,39,40]. A fully branching and space-filling porous network following Murray's law endows our material with both increased surface area for oxygen adsorption, as well as full and fast diffusion of gas molecules. The ZnO M–M–M exhibits very

high sensitivities compared to ZnO M–M with two levels of Murray regularity and ZnO M (inset in Fig. 4c). The sensitivity is further boosted with the increase of ethanol concentrations at 330 °C (Fig. 4c), showing highly enhanced gas interfacial interaction with ZnO M–M–M. The sensitivities (Supplementary Table 2) are 9 for commercial bulk ZnO, 21 for ZnO M, 40 for ZnO M–M and 93 for ZnO M–M–M at 100 p.p.m., and are 11 for ZnO bulk, 57 for ZnO M, 105 for ZnO M–M and 457 for ZnO M–M–M at 500 p.p.m., respectively. The achieved sensitivity of 457 is the highest value ever reported for ethanol detection and exceeds by at least 20 times that of commercial SnO$_2$ sensors, which have a maximum sensitivity of ~20 (refs 18,39–42). With the optimum gas diffusion and exchange network, the response speed for ZnO M–M–M is also very fast, <1 s at 100 and 250 p.p.m. (Supplementary Fig. 10). The response is over ~25 times faster than that of ZnO M, which exhibits a response time >25 s at 100 and 250 p.p.m. (Supplementary Table 2). The ZnO M–M–M also exhibits stable working performances with a high sensitivity and <1 s response/recovery time for 250 p.p.m. ethanol (Fig. 4d), and excellent retention of structure during repeated detection (Supplementary Fig. 11).

Previous approaches to promote gas-sensing performances by forming mesopores yielded limited results, including ZnO nanoplates having a sensitivity of 9 at 100 p.p.m. (ref. 39), and nest-like ZnO having a sensitivity of 74 at 500 p.p.m. (ref. 41). In addition, their response times are in the range of 10–100 s (refs 39,41). Development of high performance gas sensors by self-assembling NPs has generated enormous interest, but has yielded few advances to date, including Cu$_2$O supercrystals, showing a sensitivity of only 13 at 500 p.p.m. (ref. 18), and SnO$_2$ films with a sensitivity of 2.3 at 500 p.p.m. (ref. 42). With our Murray material having a structure similar to the breathing network of insects, we obtain gas detection over a broad concentration range with higher, faster and much more stable response compared to state-of-the art materials[39–43].

The Li-storage performances of our ZnO Murray materials self-assembled on Cu foil electrodes are presented in Fig. 4e. The discharge–charge voltage profiles of ZnO samples (Fig. 4e, inset) exhibit obvious discharge plateaus at $\sim 0.56$ V and sloping charge plateaus at $\sim 1.2$ V, consistent with the reversible reaction: $ZnO + 3Li^+ + 3e \leftrightarrow ZnLi + Li_2O$ ($978 \, mA \, h \, g^{-1}$ for $3Li^+$ per ZnO)[44]. The long discharge curves below the plateau marked by the horizontal line can be attributed to the nanoscale interfacial Li-storage, which results in larger capacity due to increased exchange of Li-ion[19,44,45]. For good cycling stability and rate capability, the Li-ion transfer process should be dominated by ion transfer in pores filled with electrolyte, rather than by sluggish solid-state ion diffusion. Thus, porous nanostructured electrodes are highly desirable[46–48]. Murray materials can meet both the criteria of fully space-filling pores, enabling ultra-short solid-phase Li-diffusion, and an optimum electrolyte-filled porous network favouring full and rapid Li-ion transfer with the electrolyte. Indeed, our ZnO M–M–M shows outstanding stability and a high reversible capacity of $\sim 1,660 \, mA \, h \, g^{-1}$ after 50 cycles at $0.05 \, A \, g^{-1}$ (Supplementary Fig. 12), along with much higher rate capabilities compared to ZnO M–M and ZnO M (Fig. 4e). Due to highly enhanced Li-ion transfer and exchange, the ZnO M–M–M exhibits capacities of 1,430, 1,260, 1,120 and $990 \, mA \, h \, g^{-1}$ at increasing current densities of 0.5, 2.5, 5 and $10 \, A \, g^{-1}$, respectively. We note that mesoporous materials self-assembled by hollow $Co_3O_4$ (ref. 19) or 3.5 nm ZnO NPs[44], have been reported with high rate capacities of $300–500 \, mAh \, g^{-1}$, however, only at a current density of $\sim 1–3 \, A \, g^{-1}$. Our hierarchical ZnO Murray material with three-level regularity can deliver a reversible capacity of $870 \, mA \, h \, g^{-1}$, even at the extremely high-current density of $20 \, A \, g^{-1}$. This is a 40-fold rate increase compared to that of ZnO M for Li-storage. Furthermore, hierarchical ZnO Murray materials exhibit a long-life cycling stability of up to 5,000 cycles with a reversible capacity of $1,260 \, mA \, h \, g^{-1}$ at a current density of $2.5 \, A \, g^{-1}$ (Fig. 4f), due to their good structural stability (Supplementary Fig. 13). The Coulombic efficiency is always 100% during the whole cycling process. The reversible capacity of $1,260 \, mA \, h \, g^{-1}$ at $2.5 \, A \, g^{-1}$ is more than 25 times higher than that of state-of-art graphite, which shows only $\sim 50 \, mA \, h \, g^{-1}$ at a similar rate ($2.5 \, A \, g^{-1}$) as a Li-ion battery anode[49]. Thus, our bio-inspired vascularized ZnO Murray network with multi-scale pores as an anode material delivers ultrahigh capacities and rate capabilities, along with long-life cycling stability.

## Discussion

Through evolution, nature has developed a number of strategies to create materials with optimized properties. Nature's designs serve as powerful sources of inspiration for us to discover and mimic material structures for various applications. Our work illustrates a bio-inspired Murray material with multi-scale macro–meso–micropores following revisited Murray's law, which offers a unique performance boost for applications benefitting from highly enhanced mass transfer and exchange. The Murray materials deliver 5-, 25- or 40-fold increases in reaction rates compared to unimodal mesoporous materials, when used as photocatalysts, gas sensors or electrodes for Li-ion batteries, respectively. We achieve ultrahigh sensitivity ($\sim 457$ with response time within 1 s), exceeding 20 times that of commercial $SnO_2$ sensors, superior rate capability ($\sim 870 \, mA \, h \, g^{-1}$ at $20 \, A \, g^{-1}$) and high reversible capacity, over 25 times higher than that of state-of-the-art anodes at a similar rate. This study demonstrates that through rational design based on Murray's Law, the performance of materials can be improved significantly and that the material design principles based on Murray's law can

achieve predictive and optimized functions. It is envisioned that our strategy is applicable to an enormous range of porous materials and has a broad scope in functional ceramics and nano-metals for energy and environmental applications. From plants, animals and materials to industrial processes, the introduction of Murray's law concept to industrial reactions can revolutionize the design of reactors with highly enhanced efficiency, minimum energy, time and raw material consumption for a sustainable future.

## Methods

**Revisiting the original Murray's law.** The derivation process is presented below (for details see Supplementary Methods)[8–10,50]. For an individual circular pore with radius of $r$ and length of $l$, the volume ($V$) of the pore is $\pi r^2 l$.

For laminar flow, $Q_1 = \frac{\pi r^4}{8 \mu} \frac{\Delta P}{l}$, where $Q_1$ is the volumetric flow rate, $\mu$ is the viscosity, and $\Delta P$ is the pressure drop across the length of the pore. And, $\frac{\partial [Q_1 \Delta P - \lambda V]}{\partial r} = 0 \Rightarrow \frac{\Delta P}{l} = \frac{1}{r} \sqrt{4 \mu \lambda}$, where $\lambda$ is the Lagrange multiplier. Thus, $Q_1 = \frac{\pi}{\sqrt{16 \mu / \lambda}} r^3 \Rightarrow Q_1 = k_1 r^3$, where $k_1$ is a constant.

For mass diffusion, $Q_2 = D \pi r^2 \frac{\Delta C}{l}$, where $Q_2$ is the amount of substance diffused through a given cross-section per unit time, $D$ is the diffusion coefficient and $\Delta C$ is the concentration difference across the pore. And, $\frac{\partial [Q_2 \Delta C - \lambda V]}{\partial r} = 0 \Rightarrow \frac{\Delta C}{l} = \sqrt{\frac{\lambda}{D}}$. Thus, $Q_2 = \pi \sqrt{\lambda D} r^2 \Rightarrow Q_2 = k_2 r^2$, where $k_2$ is a constant.

For ionic or electronic transfer, $Q_3 = \sigma \pi r^2 \frac{\Delta V}{l}$, where $Q_3$ is the electric charge transferred through a given cross-section per unit time, $\sigma$ is the conductivity and $\Delta V$ is the potential difference across the length of the pore. And, $\frac{\partial [Q_3 \Delta V - \lambda V]}{\partial r} = 0 \Rightarrow \frac{\Delta V}{l} = \sqrt{\frac{\lambda}{\sigma}}$. Thus, $Q_3 = \pi \sqrt{\lambda \sigma} r^2 \Rightarrow Q_3 = k_3 r^2$, where $k_3$ is a constant.

For connecting a parent pipe with radius of $r_0$ to many children pipes with radius of $r_i$ for mass transfer with no mass variations: $Q_0 = \sum_{i=1}^{N} Q_i$. Therefore, $r_0^\alpha = \sum_{i=1}^{N} r_i^\alpha$. For laminar flow transfer, $\alpha = 3$; For mass diffusion or ionic transfer, $\alpha = 2$.

**Deduction of the generalized Murray's law.** Murray's principle for optimizing individual pores can be considered together with other relevant conditions, and can also be applied for asymmetric and multi-branch transport networks. We deduced the generalized Murray's law for optimizing mass transfer involving mass variations in chemical reactions. Figure 2b shows a schematic image of a model porous network with a parent pipe connecting with many children pipes. For connecting a parent pipe with radius of $r_0$ to many children pipes with radius of $r_i$ for mass transfer, there could be two scenarios. First, $Q_0$ amounts of substance are imported into the parent pipe through the cross-section per unit time, with a loss of $XQ_0$ amount ($X$ is the mass loss ratio) during the transfer in parent pipe. Alternatively, $Q_0$ amounts of substance export from the parent pipe, with an associated increase by $XQ_0$ amount ($X$ is the mass increase ratio) during the transport. For endowing the optimum mass transfer with full coverage, the law of mass conservation at a junction gives: $Q_0 - XQ_0 = \sum_{i=1}^{N} Q_i$.

For optimizing the mass transfer either involving or not involving mass variations, according to Murray's optimum principle: $Q_0 = k r_0^\alpha$ and $Q_i = k r_i^\alpha$, where the exponent $\alpha$ (2 or 3) is dependent on the type of the transfer. Thus, a generalized form of Murray's law for optimizing mass transfer involving mass variations can be written: $r_0^\alpha = \frac{1}{1-X} \sum_{i=1}^{N} r_i^\alpha$, where the exponent $\alpha$ (2 or 3) is dependent on the type of the transfer, and $X$ is the ratio of mass variation during transfer in parent pore.

For example, for a porous material involving mass diffusion or ionic transfer in a chemical reaction ($\alpha = 2$): $r_0^2 = \frac{1}{1-X} \sum_{i=1}^{N} r_i^2$, because $Q_0$ is proportional to the reaction rate derived from the entire exchange surface, and $XQ_0$ is proportional to the reaction rate coming from the exchange surface of the parent pipe. As a reasonable consideration, the mass loss or increase ratio $X$ can be proportional to the surface area ratio of the parent pipe to the total pipes.

**Design of Murray materials following the generalized Murray's law.** Murray's law defines the basic geometric features for porous materials with optimum transfer properties. We used the generalized Murray's law to design and optimize the structures of hierarchically porous materials. This concept has led to materials, termed as the Murray material, whose pore sizes are multiscale and are designed with diameter ratios obeying the revisited Murray's law.

Figure 2c shows the pore model abstracted from our bio-inspired hierarchically macro–meso–microporous materials self-assembled by nanoparticles. To connect macropores with mesopores, and mesopores with micropores, the size ratios between multi-scale pores are thus derived based on Murray's law of diffusion and ion transfer networks ($\alpha = 2$). It is defined that $h$ is the height of the macropores, equating to the thickness of the film. $l$ is the length of the mesopores, equating to the average wall width for macroporous networks. $D_{macro}$ is the diameter of the macropores. $D_{meso}$ is the diameter of the mesopores. And $D_{micro}$ is the diameter of

the micropores. $S_{micro}$ is the specific surface area from the micropores, and $S$ is the specific surface area from all the NPs. $n$ is the average number of the micropores within a single NP. And $d$ is the diameter of NPs.

For connecting macropores to mesopores, the exchange surface from the macropores can be ignored relative to the whole exchange area of our macro–meso–microporous materials in chemical reactions. And the mass loss or increase ratio $X = (S_{macro})/(S_{macro} + S_{meso} + S_{micro}) \ll 1$. According to the generalized Murray's law, we can get the following equation:

$D_{macro}^2 = \sum D_{meso}^2 = \frac{\pi D_{macro}}{d} \frac{h}{d} D_{meso}^2 \Rightarrow D_{macro} = \pi h \frac{D_{meso}^2}{d^2}$.

Then, we can get the thickness of the film from the following equation:

$$h = \frac{d^2}{\pi D_{meso}^2} D_{macro} \qquad (1)$$

Connecting mesopores to micropores, the exchange surface from mesopores and micropores and the mass loss $X$ cannot be ignored, according to the generalized Murray's law we can get the following equation:

$D_{meso}^2 = \frac{1}{1-X} \sum D_{micro}^2 = \frac{1}{1-X} \frac{1}{d} n D_{micro}^2 \Rightarrow D_{micro} = [\frac{(1-X)d}{nl}]^{1/2} D_{meso}$.

As mentioned previously, the mass loss or increase ratio $X$ can be proportional to the surface area ratio of the parent pipe to all the pipes. Herein, the $X$ can be approximately proportional to the ratio of $(S - S_{micro})$ to $S$, that is, $1 - X = 1 - (S - S_{micro})/S = S_{micro}/S$.

Then, we can get the average wall width for macroporous networks from the following equation:

$$l = \frac{(1-X)dD_{meso}^2}{nD_{micro}^2} = \frac{S_{micro}}{S} \frac{dD_{meso}^2}{nD_{micro}^2} \qquad (2)$$

The physical relationships in the two equations can be used as a guide to fabricate Murray materials. When the NPs are synthesized and self-assembled, the nanoscale structural parameters for the NPs ($d$, $n$, $S$, $S_{micro}$, $D_{micro}$) and the mesopore size ($D_{meso}$) can be given and detected by TEM and argon adsorption–desorption isotherms measurements. Then, the micro-scale structural parameters including the film thickness ($h$), the wall width for macroporous networks ($l$) and the diameter of the macropores ($D_{macro}$) can be calculated from equations (1) and (2). For example, according to equation (2) with $d = 30$ nm, $n = 8$, $S_{micro}/S = 0.5$, $D_{micro} = 1.1$ nm, $D_{meso} = 18$ nm, $l$ can be calculated as $\sim 502$ nm. And according to equation (1) with $d = 30$ nm, $D_{meso} = 18$ nm, $D_{macro} = 1,000$ nm, film thickness $h$ can be calculated as $\sim 885$ nm.

**Synthesis.** The microporous ZnO NPs are synthesized in solution under an argon atmosphere. A mixture of $Zn(acac)_2$ (0.198 g, Aldrich, 99%) and oleylamine (4.01 g, Aldrich, 80%) is heated to 80 °C for 30 min with stirring. The resultant solution is heated to 150 °C and kept at this temperature for 60 min. After cooling to room temperature naturally, excess ethanol is added to the solution to give a white precipitate. This is then isolated via centrifugation. The ZnO NPs are repeatedly washed, isolated and dried, and are finally dispersed in hexane to form a suspension. Residual amines are removed via washing with ethanol, as confirmed by FTIR spectroscopy.

**Self-assembly.** The ZnO NPs are self-assembled by drop-casting after evaporation of volatile solvents on flat substrates under room-temperature and atmospheric pressure. To obtain single-layer networks for TEM observation, a hexane suspension of ZnO with concentrations varying from 0.03125 to 0.5 mg ml$^{-1}$, is drop-casted on carbon-coated copper grids placed at the centre of 20 mm × 20 mm Si wafers. A one layer porous network is formed by evaporation-driven self-assembly of microporous ZnO nanoparticle building blocks. The solution fully covers the 20 mm × 20 mm Si wafers. To prepare 3D network films (sample M–M–M) for SEM observation and further use, microporous ZnO NPs are suspended in hexane with $\sim 0.25$ mg ml$^{-1}$ concentration and are repeatedly layer-by-layer drop-casted on 20 mm × 20 mm Si wafers. ZnO M–M with two level Murray regularity is fabricated by the evaporation-driven self-assembly of microporous ZnO NPs suspended in ethanol. ZnO M is synthesized by the self-assembly of non-porous ZnO nanoparticles of 30 nm, which are synthesized using the same method as for microporous ZnO NPs but with additional reaction at 290 °C for 180 min. Thus the ZnO M sample contains only 22 nm mesoporosity without the presence of microporosity (Supplementary Table 1).

**Characterization.** Various ZnO M–M–M films are scraped off and degassed at 250 °C for 30 min and then at 120 °C for 12 h before analysis using a Micromeritics ASAP 2020 porosimeter with argon as a test gas. The specific surface area is estimated by the Brunauer–Emmett–Teller method. The pore size distribution for micropores is calculated from the desorption branch of the isotherm using the non-local density functional theory method. The pore size distribution for mesopores is calculated using the Barrett–Joyner–Halenda model. The pore size distribution for macropores is analysed using a Micromeritics AutoPore IV 9500 mercury porosimeter. The TEM images are obtained using a JEM-2100F with 200 kV accelerating voltage. The SEM images are obtained using a Hitachi S-4800. The XRD pattern is obtained using a Bruker D8 Advance diffractometer equipped with a Cu Kα rotating anode source. GISAXS data were collected on a Bruker NANOSTAR SAXS instrument equipped with a Cu Kα (1.54 Å) microfocus source,

using a high-flux pinhole set (400 µm aperture), a sample-to-detector distance of 100 cm, an analysis angle of 0.20°, and a 2,048 × 2,048 Vantec area detector.

**Photocatalysis.** The photocatalytic performance of ZnO films was evaluated via the degradation of rhodamine B (RhB) in aqueous solution under simulated solar light irradiation with a distance of 20 cm at room temperature. Different ZnO films on Si wafers are put into $10^{-5}$ M (10 ml) RhB aqueous solution and then exposed to light irradiation using a xenon lamp (PLS-SXE-300UV). The infrared spectral region is cut off by an optical filter. The incident light intensity at the location of the catalyst is 100 mW cm$^{-2}$. The photocatalytic performance is analysed by monitoring the optical absorption peak of RhB at 554 nm. RhB degradation using a Si wafer without catalyst is also carried out as a control experiment. The relative concentration is defined as the ratio $C/C_0$, where $C$ (mg l$^{-1}$) is the residual concentration of the RhB and $C_0$ (mg l$^{-1}$) is the initial concentration.

**Gas sensing.** Various ZnO films on Si wafers are scraped off and pasted onto ceramic tubes between gold electrical contacts. The gas-sensing properties of the samples are determined using a commercial instrument (Winsen WS-60A). The measurement follows a static process: a given amount of the test gas is injected into a glass chamber and mixed with air. The gas sensitivity $S$ is defined as the ratio $R_{air}/R_{gas}$, where $R_{air}$ is the electrical resistance measured in air and $R_{gas}$ is that measured in the test gas atmosphere. The response time to reach the 90% of the final equilibrium signal value represents the response performance of gas sensors.

**Lithium-ion batteries.** Various ZnO films are deposited on Cu foils using the above self-assembly methods as used on Si wafers and are finally dried at 60 °C and degassed at 250 °C under vacuum. The electrochemical performance is evaluated via a CR2032-tpye coin cell on a LAND (CT2001A) multi-channel battery test system. The batteries are fabricated using Cu foil with the ZnO film as the working electrode, a 1 M solution of $LiPF_6$ in ethylene carbon/diethyl carbonate (1:1, in wt %) is used as the electrolyte, and a lithium foil as the counter electrode. Galvanostatic discharge/charge measurements are performed over the potential range of 3–0.02 V versus $Li^+$/Li.

**Data availability.** The authors declare that the data supporting the findings of this study are available within the article and its Supplementary Information files, and all relevant data are available from the authors.

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

## Acknowledgements

B.-L.S. acknowledges the Chinese Central Government for an 'Expert of the State' position in the Program of the 'Thousand Talents' and a Life membership at Clare Hall, University of Cambridge. Y.L. acknowledges Hubei Provincial Department of Education for the 'Chutian Scholar' program. T.H. acknowledges support from the Royal Academy of Engineering (Graphlex). This work is financially supported by the National Key Research and Development Program of China (2016YFA0202602), Chinese Ministry of Education in a framework of the Changjiang Scholar Innovative Research Team Program (IRT_15R52) and the International Science & Technology Cooperation Program of China (2015DFE52870). This work is also supported by the project supported by State Key Laboratory of Advanced Technology for Materials Synthesis and Processing (Wuhan University of Technology), and Shenzhen Basic Research Program (No. JCYJ20160422091418366). C.J.B. and D.D. acknowledge support from the Department of Energy Office of Science Basic Energy Sciences Catalysis and Materials Science and Engineering programs, the Air Force Office of Scientific Research, and the Sandia National Laboratories Lab Directed R&D program.

## Author contributions

B.-L.S., X.Z. and Y.L. conceived the idea of the project. X.Z. performed the formula deduction and theoretical calculation for designing materials. X.Z. and G.S. devised and performed the processes for materials synthesis and self-assembly, and characterized the performances of materials. Y.L., X.Z. and C.W. carried out the TEM and SEM observation. D.D. carried out the GISXAS measurement. X.Z. wrote the draft manuscript. Y.L., T.H., C.J.B. and B.-L.S. revised the manuscript. B.-L.S. supervised the project, discussed and cooperated with C.J.B. to interpret the joint experiments, modelling and theoretical studies and finalized the manuscript.

## Additional information

**Competing interests:** The authors declare no competing financial interests.

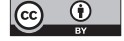

