## [Peer review file · Nature Communications]

Reviewers' comments:

Reviewer #1 (Remarks to the Author):

The paper reports the use of Murray's principle for the development of hierarchical porous ZnO materials with improved properties in mass transfer and ion transport. The concept of Murray's principle is rarely explored and materials science and, thus, the example given here could stimulate a new research field among hierarchical porous materials. Therefore, I feel that the topic is novel, interesting and in principle suitable for Nature Communications.

Nevertheless, few points should be addressed prior to acceptance of the article:

- The title is somewhat fragmented! What is transferred? Is it mass transfer and ion transport?
- The diameter of the micropores was estimated to be 1.1 nm. I am not convinced that the HK method is suitable here to give reliable pore sizes from nitrogen isotherms. I strongly suggest to employ argon as a test gas and NLDFT for calculating the pore size.
- The observed differences in the catalytic activity are explained basically by means of an optimized mass transport of the reactant. This argument should be strengthened by reporting diffusion measurements e.g. in the form of adsorption kinetics. Moreover, the amount of accessible zinc atoms should be estimated.
- In the gas sensing example, the ZnO M-M sample appears to perform better than the Murray Material (see e.g. Figure S7). Please comment.
- The general description of Murray's law in the method section should be moved to the supporting information

Reviewer #2 (Remarks to the Author):

This manuscript synthesized ZnO M-M-M films coincide with Murray's law through a layer-by-layer evaporation-driven self-assembly technology. The ZnO M-M-M films exhibited good performance in photocatalysis, gas sensing and as Li-ion battery electrodes. However, at this stage, this manuscript fail to provide clear evidence of significant scientific advance and new physical insight into the area of applied materials and chemistry. As a typical flagship journal, the work submitted to it should represent a clear conceptual and methodological advance that would likely generate interest among our readership. As such, the manuscript is more suitable for publication in a specialized journal.

Reviewer #3 (Remarks to the Author):

Murray's law is a description for optimized transfer and exchange performance in biological system. The author used the principle into the design of materials cleverly. The understanding of Murray's law is full and deep enough. The property of the hierarchical macro-meso-microporous ZnO in three applications is obvious effective. I think this is an interesting work and can be accepted by Nature Communications. However, there still some questions confused me. The authors may give distinct answers.

Q1. There are already many work about synthesis of hierarchical pores materials such as hierarchical zeolite. What is the difference between your strategy and traditional one?

Q2. Hierarchical pores of biological networks are integral structure. The hierarchically porous networks of your materials are formed by Evaporation-Driven Self-Assembly strategy. So I think the equivalence of these two system is not very certain.

Q3. I suggest that your building block ZnO nanoparticles could not be called monodisperse based on your TEM images. The standard of monodisperse require almost identical in size and

morphology.

Q4. How could you identify that the pores structure is main factor for ZnO in degradation of RhB? More descriptions and references about the influence of pores on catalytic activity should be added.

Q5. Whether the pores morphology bring impacts on the property of surface or energy band gap of the ZnO, which may result in enhanced performance.

Q6. The authors may revise images of TEM or HRTEM to see the micropores more clearly.

Q7. It's not certain whether your microporous ZnO nanoparticles is constructed by smaller ZnO nanoparticles or single component through your characterization.

Q8. Is oleylamine the surfactant of all ZnO nanomaterials? Though the surfactant always bring negative effects to the catalytic activity, it's necessary to take the surface difference of your ZnO and bulk into account.

Q9. I suggest that the authors may consider the performance of surface-clean hierarchical macro-meso-microporous ZnO. On the other hand, the process can also be used to investigate the thermal stability.

Q10. Brief introduction of the importance of the three applications should be added to manifest the practicability of your materials.

Q11. The authors may refer more recent publications: (A method for modeling and deciphering the persistent photoconductance and long-term charge storage of ZnO nanorod arrays, Nano Research 2016, 9, 2972; A facile synthesis of hierarchical Sn₃O₄ nanostructures in an acidic aqueous solution and their strong visiblelight- driven photocatalytic activity, Nano Research 2015, 8, 3553; ZnO nanostructures in enzyme biosensors; Hierarchical carbon nanocages as high-rate anodes for Li- and Na-ion batteries, Nano Research 2015, 8, 3535)

Response to Reviewers' report

To Reviewer 1

We thank you for your positive comments as well as inputs and suggestions, which are very helpful in improving the quality of our manuscript. Please find below our answers to your questions and suggestions. We have modified the manuscript by carefully considering each of your comments.

The paper reports the use of Murray's principle for the development of hierarchical porous ZnO materials with improved properties in mass transfer and ion transport. The concept of Murray's principle is rarely explored and materials science and, thus, the example given here could stimulate a new research field among hierarchical porous materials. Therefore, I feel that the topic is novel, interesting and in principle suitable for Nature Communications. Nevertheless, few points should be addressed prior to acceptance of the article:

1. The title is somewhat fragmented! What is transferred? Is it mass transfer and ion transport?

Thank you for this very valuable suggestion. Murray's principle concerns mass transfer. Therefore, we adopted a more accurate title "Bio-inspired Murray materials for unprecedented mass transfer and activity".

2. The diameter of the micropores was estimated to be 1.1 nm. I am not convinced that the HK method is suitable here to give reliable pore sizes from nitrogen isotherms. I strongly suggest to employ argon as a test gas and NLDFT for calculating the pore size.

Thank you again for this very important suggestion. Nitrogen has been traditionally used as a probe molecule for the determination of the pore diameters of materials. For micropore size estimation, for example, for zeolites, HK method is generally used. However, argon has been recently proposed as an alternative adsorbate for more reliable determination of the porous structures because of its smaller size and no dipole or quadrupole moment. Isotherms supplied now are made using Ar as test gas. As per your suggestion, we have now used the NLDFT method for calculating the micropore size instead of HK method in the revised manuscript. It shows that the sharpness of the peak has marginally increased, but the location of the peak is still centered at around 1.1 nm. We have updated this result in Figure 3I in the revised manuscript. We have also corrected a typo and revised the information on adsorption-desorption measurements using argon as test gas in the Methods section. Please find the changes at the corresponding places in the revised manuscript (in red).

3. The observed differences in the catalytic activity are explained basically by means of an optimized mass transport of the reactant. This argument should be strengthened by reporting diffusion measurements e.g. in the form of adsorption kinetics. Moreover, the amount of accessible zinc atoms should be estimated.

Thank you for this very valuable comment. Following your suggestion, we have added more descriptions and references and carried out the RhB adsorption kinetics measurements to strengthen our argument on the influence of pores on mass transfer and photocatalytic activity in the revised manuscript. The adsorption kinetics in dark of RhB has been now included in Supplementary Figure 9. Introducing macro-mesopores in photocatalyst could indeed promote mass transfer and degradation rate. We compared ZnO M-M-M (Macro-meso-micropores, three level Murray regularity) and ZnO M-M (Meso-micropores, two level Murray regularity) with equivalent contact surface area ($76 \text{ m}^2/\text{g}$, Supplementary Table 1). Compared to ZnO M-M without macropores, the presence of macropores connecting to mesopores following Murray's Law in ZnO M-M-M increases the degradation rate by 2.5 fold, which can be clearly attributed to a boosted mass transfer in macro-meso-microporous Murray networks.

Highly enhanced accessibility of the diffusing species inside of ZnO-M-M-M materials was further confirmed by monitoring the adsorption kinetics of RhB in dark following a pseudo-first-order model, revealing that ZnO M-M-M delivered a 2.5 fold faster adsorption rate of RhB than ZnO M-M (seen in the figure below, and Supplementary Figure 9 in the revised Supplementary Information). The residual concentration of the RhB at equilibrium is 0.75 for ZnO M-M-M, 0.82 for ZnO M-M, and 0.825 for ZnO M, respectively. Thus, we can estimate the amount of accessible Zn atoms from the adsorption measurement. The concentration of RhB aqueous solution is 10^{-5} M (10 mL) in our experiment. Based on the adsorption modes of RhB (two zinc atoms for adsorbing one RhB molecule) (Qi Wang, *Langmuir* 2008, 24, 7338-7345), the amount of accessible zinc atoms for adsorbing RhB is estimated to be $3.1 \times 10^{-5} \text{ mol/g}$ for ZnO M-M-M, $2.3 \times 10^{-5} \text{ mol/g}$ for ZnO M-M, and $2.2 \times 10^{-5} \text{ mol/g}$ for ZnO M, respectively.

Please find them at the corresponding place in the revised manuscript and revised supplementary Information highlighted in red colour.

Supplementary Figure 9 | Adsorption in dark of RhB for different ZnO samples. Before the adsorption measurement, the samples are repeatedly washed using distilled water. a) Variation of relative concentration of RhB (C/C_0) vs time (t) for different ZnO samples. The relative concentration is defined as the ratio C/C_0 , where C (mg/l) is the residual concentration of the RhB and C_0 (mg/l) is the initial concentration of the RhB. b) Pseudo-first order plots of RhB adsorption in dark on different ZnO samples as a function of adsorption time. The pseudo-first-order model for adsorption of RhB is depicted as: $\ln(q_e - q) = \ln(q_e) - kt/2.303$, where q_e (mg/g) and q (mg/g) are the adsorption capacity of RhB onto ZnO at equilibrium and at time t , respectively. And the term k (1/min) is the pseudo-first order rate constant, which can be obtained from the slope of plots of $-\ln[(q_e - q)/q_e]$ versus time t . Also, $q = (C_0 - C)V/M$, where V is the solution volume (L), and M is the mass of the adsorbent (mg). It reveals that the adsorption rate of ZnO M-M-M is 2.5 times higher than that of ZnO M-M or ZnO M.

4. In the gas sensing example, the ZnO M-M sample appears to perform better than the Murray Material (see e.g. Figure S7). Please comment.

Thank you for this comment. In the main text, Figure 4c (see the left figure below) shows the sensitivities of different ZnO samples exposed to ethanol vapor of various concentrations. The gas sensitivity S is defined as the ratio R_{air}/R_{gas} , where R_{air} is the electrical resistance measured in air and R_{gas} is that measured in the test gas atmosphere (Kim et al, J. Am. Chem. Soc., 2016, 138, 13431-13437). To improve clarity, we have now added a new table (Supplementary Table 2). It summarizes the sensitivity measured for a series of samples. This also reveals that the sensitivities of ZnO M-M are 40 at 100 ppm, 93 at 250 ppm, and 105 at 500 ppm while the sensitivities of ZnO M-M-M are 93 at 100 ppm, 419 at 250 ppm, and even 457 at 500 ppm. Therefore, the ZnO M-M-M sample with three level Murray regularity demonstrated much higher sensitivities than that of ZnO M-M with two level Murray regularity at various ethanol concentrations.

In the supplementary Figure 10 (see the right figure below, formerly Figure S7) shows the response performance of different ZnO samples exposed to ethanol vapor with various concentrations. The response time to reach the 90% of the final equilibrium signal of voltage is usually used to represent the response performance of gas-sensors (Jing, Z., et al, *Adv. Mater.* 20, 4547-4551, 2008; Lee, J., et al, *Sensor. Actuat. B* 140, 319-336, 2009; Xu, X., et al, *J. Am. Chem. Soc.* 130, 12527-12535, 2008.). The Y-axis (right figure) represents the voltage changes. The content of voltage is not indicative of response performance or sensitivity since the content of voltage depends also on the sample amount. However, by measuring voltage variation, we can determine the response performance (time). The shorter time to reach equilibrium, the higher the response performance. The Supplementary Table 2 shows all the measurements.

From Supplementary Figure 10 and Supplementary Table 2, it is clear that the response time to reach the 90% of the equilibrium value for ZnO Bulk is very long from 50 to 500 ppm, showing the very low sensitivity and poor response performance. ZnO M exhibits a response time of 35s, 30s and 26s at 50, 100 and 250 ppm, respectively. ZnO M-M exhibits a response time of 30s and 26s at 50, and 100 ppm, respectively. However, the response for ZnO M-M-M is very fast, within 5 s at 50 ppm, < 1 s at 100-500 ppm. Therefore, ZnO M-M-M sample with three level Murray regularity demonstrated faster response speed than that of ZnO M-M. The response speed of ZnO M-M sample with two level Murray regularity is much higher than that of ZnO M with the presence of only mesopores. We have modified Figure S7 and supplied new Figure sets (Supplementary Figure 10, formerly Figure S7) and a new Table (Supplementary Table 2) showing more clearly the performance of different samples in ethanol sensing. Please find them at the corresponding place in the manuscript highlighted in red colour.

Sample	Gas sensitivity (R_{air}/R_{gas})			Response time (s)		
	100 ppm	250 ppm	500 ppm	50 ppm	100 ppm	250 ppm
ZnO M-M-M	93	419	457	5	< 1	< 1
ZnO M-M	40	93	105	30	26	5

ZnO M	21	51	57	35	30	26
Bulk ZnO	9	11	11	50	32	30

Supplementary Table 2 | Gas sensitivity and response speed of ZnO samples. The sensitivity is defined as the ratio $R_{\text{air}}/R_{\text{gas}}$, where R_{air} is the electrical resistance measured in air and R_{gas} is that measured in the test gas atmosphere. The response time to reach the 90% of the final equilibrium signal of voltage is used to represent the response speed of gas-sensors.

5. The general description of Murray's law in the method section should be moved to the supporting information

Thank you for this comment. According to the "Guide to Authors" of Nature Communications, The Methods section should be written as concisely as possible but should contain all elements necessary to allow interpretation and replication of the results. This section should be subdivided by short headings referring to methods used and should typically not exceed 3,000 words but may be longer if necessary". **The section "Methods" is placed at the end of the main text.**

Our "Methods" section includes about 2300 words. In this part, we gave the general description of the Murray's principle, the generalized Murray's law we developed specifically for the design of bioinspired vascularized materials and different relationship on the basis of Murray's Law between macropore size, mesopore size, micropore size, wall thickness and layer thickness of materials. This part contains all elements necessary to allow interpretation and replication of the results. The characterization techniques used are also as usual presented in this part.

Following the "Guide to Authors" of the "Nature Communications", we kept "Methods" section at the end of main text since we believe it will allow interpretation and replication of our results. We would like to thank you for your understanding.

Finally, we would like to thank you for your professionalism, positive comments and valuable suggestions to improve the quality of our manuscript.

To Reviewer 2

We are very grateful for your input and suggestions, which are indeed very helpful in improving the quality of our manuscript. Please find below our answers to your questions and suggestions. We have now modified the manuscript to provide clear evidence of significant scientific advance, new physical insight into the area of applied materials and chemistry and clear conceptual and methodological advance. We hope that our efforts in addressing your questions and comments in detail and the addition of more information and discussion have significantly improved the quality of our manuscript and satisfy your concerns.

This manuscript synthesized ZnO M-M-M films coincide with Murray's law through a layer-by-layer evaporation-driven self-assembly technology. The ZnO M-M-M films exhibited good performance in photocatalysis, gas sensing and as Li-ion battery electrodes.

However, at this stage, this manuscript fail to provide clear evidence of significant scientific advance and new physical insight into the area of applied materials and chemistry. As a typical flagship journal, the work submitted to it should represent a clear conceptual and methodological advance that would likely generate interest among our readership.

Thank you for this comment.

At first we would like to emphasize that there are no papers in the literature, which have employed Murray principles for materials design and fabrication. Our work is the first of this kind.

Clear evidence of significant scientific advance

Materials have long been the engine of technological advancement. However, developing a science of synthesis that enables predictable, controllable production of hierarchical materials with structural features optimized on multiple length scales is critical to realize the vision of “material-properties-by-design”, since no such general synthesis guidelines exist. Is it possible to establish such material design principles to achieve predictive, optimized functions? **This work, for the first time, demonstrates the synthesis of simultaneously optimized multi-length scale materials following design rules we developed on the basis of Murray principals that evolved in natural hierarchical systems and that enable the functionality of materials networks to be predictably controlled.**

Clear conceptual and methodological advance

Both plants and animals possess analogous tissues containing hierarchical networks of pores with pore size ratios that evolved to maximize mass transport and rates of reactions. The underlying physical principles of this optimized hierarchical design are embodied in Murray's law. To date we have not realized the benefit of mimicking nature's Murray networks in synthetic materials due to the difficulty in fabrication of 'vascularized' materials composed of interconnected channels with precise dimensions spanning the macro, meso, and micro length scales. **By mimicking nature's hierarchically structured networks in synthetic materials with multi-scale pores based on Murray's law, such synthetic Murray materials can potentially offer unprecedented structural superiority and performance enhancement for a wide range of applications.**

The original Murray's law is only applicable for transfer processes with no mass variations. In our work, **we revisited the original Murray's principle and further developed, for the first time, the formula of generalized Murray's law.** This is widely applicable for chemical reaction and other mass diffusion or ionic transfer processes involving mass variations. These efforts with our further theoretical work allowed us to take the first step to use the Murray's principle in the design of materials **networks with predictable and controlled functionality.**

Murray derived his law for optimal cardiovascular design. Since its discovery as the basis for vascular and blood systems, Murray's law attracted very little attention, and has thus far been completely overlooked in the field of physics, chemistry and materials.

The original Murray's law ($r_0^\alpha = \sum_{i=1}^N r_i^\alpha$) was a description for optimizing transfer process involving no mass variations, which could not be applicable for chemical reactions involving mass variations.

In this manuscript, we further revisited Murray's principle. Significantly, we deduced the formula of generalized Murray's law ($r_0^\alpha = \frac{1}{1-X} \sum_{i=1}^N r_i^\alpha$), which is widely applicable for chemical reaction and other mass diffusion or ionic transfer processes involving mass variations.

Moreover, we took the first step to bring this scientific insight into the design of materials applied for chemical applications, and established two physical formulas to direct us to construct Murray materials networks.

$$h = \frac{d^2}{\pi D_{meso}^2} D_{macro} \quad (1)$$

$$l = \frac{(1-X)dD_{meso}^2}{nD_{micro}^2} = \frac{S_{micro}}{S} \frac{dD_{meso}^2}{nD_{micro}^2} \quad (2)$$

This new methodological advance: generalized Murray's law and two formulas has been used to design materials and has led to a new concept of "Murray materials", as a new kind of material possessing multi-scale pores with diameter-ratios obeying Murray's law as illustrated in the image below.

Another new methodological advance to construct Murray material networks with different levels of regularity we made is the development of a new Layer-By-Layer, Bottom-Up, Evaporation-Driven Self-Assembly of microporous nanocrystal 'building blocks' under ambient conditions. This new synthesis method would offer a very broad and generalized approach towards the development of Murray materials.

New physical insight into the area of applied materials and chemistry

The above new scientific concept and clear conceptual and methodological advance are immediately applicable. We show **the first example of the design and synthesis of materials following Murray's law that incorporate macro-meso-micropores in one solid body**, via a bottom-up approach on the basis of a completely new Layer-by-Layer, Evaporation Driven Self-Assembly Process (EDSA) using well defined and monosized microporous ZnO nanoparticles as the primary building blocks. **Such unprecedented bio-inspired vascularized materials whose pore sizes regularly decrease across multiple scales and finally terminate in size-invariant units like plant stems, leaf veins and respiratory systems provide hierarchical branching and precise diameter-ratios for connecting multi-scale pores from macro to micro levels and**

may enable highly enhanced mass exchange and transfer in liquid-solid, gas-solid and electrochemical reactions.

Indeed, the materials synthesized exhibit unprecedented performance in various applications. Full comparison to the state-of-the art shows the clear superiority of such bio-inspired Murray materials. For example, our fabricated Murray materials deliver 5-, 25-, or 40-fold increases in reaction rates compared to unimodal mesoporous materials, when used as photocatalysts, gas sensors, or electrodes for Li-ion batteries, respectively. We achieve ultrahigh sensitivity $R_{\text{air}}/R_{\text{gas}}$ (~ 457 with response time within 1s), exceeding 20 times that of commercial SnO₂ sensors, superior rate capability ($\sim 870 \text{ mA h g}^{-1}$ at 20 A g^{-1}) and high reversible capacity, over 25 times higher than that of state-of-the-art anodes at a similar rate.

This study demonstrated that only by rational design based on Murray's Law, the performance of materials can be improved significantly and that the material design principles based on Murray's law can achieve predictive and optimized functions.

Therefore, our present work from the special physiological Murray's principle, to the generalized Murray's law for designing Murray materials, and to the practical synthesis and application of Murray materials essentially represents clear evidence of significant scientific advance and brings new physical insight into the area of applied materials and chemistry.

Scientific and technological Impact

Using the generalized Murray's law to design materials with boosted performances for energy and environmental applications can pave the way for pursuing optimized properties of porous materials for various applications.

It is envisioned that our strategy is applicable to an enormous range of porous materials, and has a broad scope in functional ceramics and nano-metals for energy and environmental applications. From plants, animals and materials to industrial processes, the introduction of the Murray's law concept to industrial reactions can revolutionize the design of reactors with highly enhanced efficiency, minimum energy, time, and raw materials consumption and unprecedented sustainability.

Indeed, Reviewer 1 has suggested that this work could stimulate a new research field. We also strongly believe that this work will interest the broadest possible readership of Nature Communications working in the field of physics, chemistry, mathematics, biology, materials science and engineering, energy and environmental science and technology, and represents a significant breakthrough that is of general

interest.

In the revised manuscript, we have strengthened and clarified the evidence of significant scientific advance and new physical insight in different areas. Please find them at the corresponding places in the revised manuscript highlighted in red colour.

Finally, we would like to thank you again for your time and effort to evaluate our work and comments to help us to strengthen the manuscript on clear evidence of significant scientific advance, new physical insight in the area of applied materials and chemistry and clear conceptual and methodological advance. We believe that this improvement will be highly beneficial to the broad readership of Nature Communications.

To Reviewer 3

We are very grateful for your input and suggestions, which are indeed very helpful in improving the quality of our manuscript. Thank you very much for your positive and encouraging comments. Please find below our answers to your questions and suggestions. We have modified the manuscript by carefully considering each of your comments.

Murray's law is a description for optimized transfer and exchange performance in biological system. The author used the principle into the design of materials cleverly. The understanding of Murray's law is full and deep enough. The property of the hierarchical macro-meso-microporous ZnO in three applications is obvious effective. I think this is an interesting work and can be accepted by Nature Communications. However, there still some questions confused me. The authors may give distinct answers.

1. There are already many works about synthesis of hierarchical pores materials such as hierarchical zeolite. What is the difference between your strategy and traditional one?

Thank you for this very valuable comment.

The main differences between our strategy and traditional one are summarized below:

- 1) Our approach is based on the Murray's Law to design bio-inspired vascularized materials with well-defined porous hierarchy. There exists a direct relationship between macropore, mesopore and micropore sizes. This strategy leads to the formation of materials with hierarchical branching and precise diameter-ratios for connecting multi-scale pores from macro to micro levels.

On the basis of our long experience in the synthesis of hierarchically porous materials, conventional preparation methods can not lead to the simultaneous control of pore sizes at different length scales in materials although sometimes one or two different pore sizes can still be tailored. In materials synthesized using such conventional methods, the relationship between different pore sizes does not exist and obtained pores are not necessarily interconnected.

- 2) Conventional approach to fabricate ordered macropores uses generally self-assembled silica or polymer spheres as a starting template. Further removal of sacrificial templates by washing or calcination at high temperature severely restricts the choice of materials and thus, functionality.

For the construction of hierarchically vascularized porous materials with different level of Murray regularity and taking inspiration from nature in self-organization of cell units into complex organisms, we developed a Template-free Bottom-up Layer-by-Layer Evaporation-Driven Self-Assembly strategy which therefore can be applicable to an enormous range of Murray materials with pore size control.

We have added a comment on the progress of synthesis strategy in the introduction of the revised manuscript. Please find it at the corresponding place in the manuscript highlighted in red colour.

2. Hierarchical pores of biological networks are integral structure. The hierarchically porous networks of your materials are formed by Evaporation-Driven Self-Assembly strategy. So I think the equivalence of these two system is not very certain.

This is a very good point. Thank you. The synthesis method we developed allowed us to synthesize bio-inspired vascularized materials network with very similar features of natural Murray materials. **Our unprecedented bio-inspired vascularized materials whose pore sizes regularly decrease across multiple scales and finally terminate in size-invariant units like plant stems, leaf veins and respiratory systems provide hierarchical branching and precise diameter-ratios for connecting multi-scale pores from macro to micro levels. The adsorption-desorption results, HRTEM and SEM images and GISAXS measurements confirm such features of natural materials in our synthetic materials.**

There are also many other analogous facets between the biological Murray networks and our bio-inspired Murray materials. Firstly, the biological networks were self-organized by the basic unit of functional cell. Our Murray materials were self-assembled using the functional nanoparticles as basic building blocks. Secondly, the biological Murray networks such as the leaf vein usually built by the layer-by-layer porous networks shown in the Figures 1b (see in the top image below). Our Murray materials were also constructed by layer-by-layer structures shown in the Figures 3a, b and e (see in the bottom image below).

We have strengthened this bio-inspired strategy in the introduction of the revised manuscript. Please find it at the corresponding place in the manuscript highlighted in red colour.

3. I suggest that your building block ZnO nanoparticles could not be called monodisperse based on your TEM images. The standard of monodisperse require almost identical in size and morphology.

Thank you for this very valuable suggestion. We have deleted the monodisperse description in the revised manuscript.

4. How could you identify that the pores structure is main factor for ZnO in degradation of RhB? More descriptions and references about the influence of pores on catalytic activity should be added.

Thank you for this very valuable comment. Following your suggestion, we have added more descriptions and references in the revised manuscript.

To identify the main impact of the pore structure on mass transfer and photocatalytic activity for ZnO in degradation of RhB and to confirm highly enhanced accessibility of the diffusing species inside ZnO-M-M-M materials, we compared ZnO M-M-M (Macro-meso-micropores, three level Murray regularity) and ZnO M-M (Meso-micropores, two level Murray regularity) with equivalent contact surface area ($76 \text{ m}^2/\text{g}$, Supplementary Table 1). The adsorption kinetics in dark of RhB following a pseudo-first-order model was monitored. This study reveals that ZnO M-M-M delivered a 2.5 fold faster adsorption rate of RhB than ZnO M-M (see the figure below, Supplementary Figure 9 in revised Supplementary Information).

Concerning photocatalytic degradation of RhB, compared to ZnO M-M without macropores, the presence of macropores connecting to mesopores following Murray Law in ZnO M-M-M increases the degradation rate by 2.5 fold (Figure 4a). This can be clearly attributed to a boosted mass transfer in macro-meso-microporous Murray networks. Introducing macro-mesopores with Murray regularity in photocatalyst could indeed promote mass transfer and degradation rate. Based on the clear experimental evidence, we state that the pore structure is main factor for ZnO in degradation of RhB.

Please find the changes at the corresponding place in the revised manuscript and revised supplementary Information highlighted in red colour.

Supplementary Figure 9 | Adsorption in dark of RhB for different ZnO samples. Before the adsorption measurement, the samples are repeatedly washed using distilled water. a) Variation of relative concentration of RhB (C/C_0) vs time (t) for different ZnO samples. The relative concentration is defined as the ratio C/C_0 , where C (mg/l) is the residual concentration of the RhB and C_0 (mg/l) is the initial concentration of the RhB. b) Pseudo-first order plots of RhB adsorption in dark on different ZnO samples as a function of adsorption time. The pseudo-first-order model for adsorption of RhB is depicted as: $\ln(q_e - q) = \ln(q_e) - kt/2.303$, where q_e (mg/g) and q (mg/g) are the adsorption capacity of RhB onto ZnO at equilibrium and at time t , respectively. And the term k (1/min) is the pseudo-first order rate constant, which can be obtained from the slope of plots of $-\ln[(q_e - q)/q_e]$ versus time t . Also, $q = (C_0 - C)V/M$, where V the solution volume (L), and M the mass of the adsorbent (mg). It reveals that the adsorption rate of ZnO M-M-M is 2.5 times higher than that of ZnO M-M or ZnO M.

5. Whether the pores morphology bring impacts on the property of surface or energy band gap of the ZnO, which may result in enhanced performance.

Thank you for this comment. Following your suggestion, we have carried out UV-vis absorption measurement of the as-synthesized ZnO nanoparticles with or without the micropores (please see below). We have not found any obvious changes for the absorption peak in the UV-vis absorption spectra. We have

added this result as the Supplementary Figure 8 in the revised manuscript. In the present study, we did not observe significant impacts of the pore morphology to the property of surface or energy band gap of the ZnO as we used the same microporous ZnO nanocrystals as the building block to construct our bio-inspired materials networks. Please find them at the corresponding place in the manuscript highlighted in red colour.

6. The authors may revise images of TEM or HRTEM to see the micropores more clearly.

Thank you for this very valuable suggestion. We have upgraded HRTEM images in the Supplementary Information. In the Supplementary Figures 1 and 2, now we can see the micropores more clearly. Please find them at the corresponding place in the manuscript highlighted in red color. Please see also more details in the next response.

7. It's not certain whether your microporous ZnO nanoparticles is constructed by smaller ZnO nanoparticles or single component through your characterization.

Thank you for this very valuable suggestion. From newly updated Supplementary Figures 1 and 2, we have observed the lattice fringes around micropores within ZnO nanoparticles through HRTEM. It revealed that the microporous ZnO was crystalline and that the crystallographic orientation of ZnO around the micropores was well in same line (see the left image below). Therefore, the nanocrystal was not constructed by smaller ZnO nanoparticles. It was a single nanocrystal component with many open microporous channels passing through it (see the right image below), which formed by the fast release of the gas molecules during the formation of nanocrystal. We have added these images and discussions as

Supplementary Figures 1 and 2 in the revised manuscript. Please find them at the corresponding place in the manuscript highlighted in red colour.

8. Is oleylamine the surfactant of all ZnO nanomaterials? Though the surfactant always bring negative effects to the catalytic activity, it's necessary to take the surface difference of your ZnO and bulk into account.

Thank you for this very valuable comment. In our experiment, oleylamine was used as surfactant to synthesize ZnO nanoparticles. Indeed, the oleylamine surfactant always bring negative effects to the catalytic activity. Thus, the residual amine on the surface of the ZnO nanoparticles has been completely removed by full and repeated washing with ethanol in our experiment. The complete removal of surfactant was checked by the FTIR spectroscopy. We then used the obtained ZnO microporous nanocrystals to self-assemble our Murray Materials with different level regularity. Therefore, we did not consider the surface difference in the manuscript. We have already introduced this in the methods section of the revised manuscript.

9. I suggest that the authors may consider the performance of surface-clean hierarchical macro-meso-microporous ZnO. On the other hand, the process can also be used to investigate the thermal stability.

Thank you for this comment. As discussed above, the residual amine on the surface of the ZnO nanoparticles has been completely removed by full and repeated washing with ethanol in our experiment. Therefore, our hierarchical macro-meso-microporous ZnO are surface-clean. On the other hand, our ZnO nanoparticles have good thermal stability. During the gas-sensing experiment, the sensitivities of the samples were measured at 330 °C. Owing to the good thermal stability for crystalline micropores and high-melting point for oxide nanoparticles in 3D compact networks, the micro-meso-macroporous structure is well retained with stable and reproducible response during the gas-sensing detections at this high temperature.

10. Brief introduction of the importance of the three applications should be added to manifest the practicability of your materials.

Thank you for this very valuable suggestion. We have added a brief introduction of the importance of the applications in photocatalysis, gas-sensing and Li-storage in the revised manuscript. Please find them at the corresponding place in the manuscript highlighted in red colour.

11. The authors may refer more recent publications: (A method for modeling and deciphering the persistent photoconductance and long-term charge storage of ZnO nanorod arrays, Nano Research 2016, 9, 2972; A facile synthesis of hierarchical Sn3O4 nanostructures in an acidic aqueous solution and their strong visiblelight- driven photocatalytic activity, Nano Research 2015, 8, 3553; ZnO nanostructures in enzyme biosensors; Hierarchical carbon nanocages as high-rate anodes for Li- and Na-ion batteries, Nano Research 2015, 8, 3535)

Thank you for this very valuable suggestion. We have added these recent references in the revised manuscript. Please find them at the corresponding place in the manuscript highlighted in red colour.

Finally, on behalf of all the authors, I would like to thank you for your very valid and helpful comments and suggestions, which have greatly helped us to improve our manuscript for Nature Communications.

REVIEWERS' COMMENTS:

Reviewer #1 (Remarks to the Author):

The authors have clearly improved the paper by responding in a detailed manner to the concerns raised by the referees. The vast majority of the suggestions were taken up and remaining questions were satisfactorily resolved. Thus, I recommend to accept the paper for publication in Nature Commun.

Some minor points:

Typo on page 9 in the heading of the paragraph: Fabrication and not Frabrication

Page 15: supplementary instead of supplemenray
response speed: better response time?

I still feel that the method section with respect to Murray's law is too long and does not provide any new information beyond literature knowledge.

Martin Hartmann

Reviewer #3 (Remarks to the Author):

The revised manuscript can be accepted now.

Response to Reviewers' report

To Reviewer 1

We thank you for your positive comments as well as inputs and suggestions. We have modified the manuscript by carefully considering each of your comments.

The authors have clearly improved the paper by responding in a detailed manner to the concerns raised by the referees. The vast majority of the suggestions were taken up and remaining questions were satisfactorily resolved. Thus, I recommend to accept the paper for publication in Nature Commun.

Some minor points:

1. Typo on page 9 in the heading of the paragraph: Fabrication and not Frabrication

Thank you for your help. We have corrected this word.

2. Page 15: supplementary instead of supplemenray

Thank you for your help. This has been corrected.

3. response speed: better response time?

Thank you for your remark. Yes, you are right, it indicates better or shorter response time. In the literature, "response speed" is indeed most commonly used to describe one of key performance parameters of a sensor.

4. I still feel that the method section with respect to Murray's law is too long and does not provide any new information beyond literature knowledge.

Thank you for this very valuable suggestion. We now have moved a large part of the description of the original Murray's Law to the Supplementary Methods. We have kept only the part concerning how we revisited the original Murray's Law to design vascularized structures providing hierarchical branching and precise diameter-ratios for connecting multi-scale pores from the macro to micro levels.

Finally, we would like to thank you for your professionalism, positive comments and valuable suggestions to improve the quality of our manuscript.

To Reviewer 3

1. The revised manuscript can be accepted now.

Thank you very much for your positive recommendation.